

# Hydroclimatic Variability and Weather-Type Characteristics in the Levant During the Last Interglacial

Efraim Bril[1], Adi Torfstein[1,2], Roy Yaniv[2,3], and Assaf Hochman[1]

[1]Fredy and Nadine Herrmann Institute of Earth Sciences, The Hebrew University of Jerusalem,  Jerusalem, Israel
[2]Interuniversity Institute for Marine Sciences, Eilat, 88103, Israel
[3]Center for Climate Medicine and human health, Gertner Institute, Sheba medical center, Derech Sheba 2, Ramat Gan, Israel

**Correspondence:** Efraim Bril (efraim.bril@mail.huji.ac.il), Adi Torfstein (adi.torf@mail.huji.ac.il), and Assaf Hochman (Assaf.Hochman@mail.huji.ac.il)

**Abstract.** Proxy-based reconstructions of the Last Interglacial peak indicate changes in precipitation characteristics in the Levant. These reconstructions suggest that precipitation occurred in brief and intense events, particularly in the region's southern parts. Some studies have offered conflicting paradigms for explaining hydroclimate variability. However, these have yet to be consistently tested in a modeling framework. Indeed, the modeling approach can undoubtedly enhance the combined

interpretation of proxy records and our understanding of hydroclimate processes in the past. We used simulations from the Paleoclimate Model Intercomparison Project 4th phase (PMIP4) to evaluate and reconstruct the precipitation characteristics of the Levant. First, we identified the Alfred Wagner Institute Earth System Model to largely resemble proxy reconstructions. Then we used it to understand the variability of hydroclimate. We examined changes in the frequency, seasonality, and persistence of the Levant's rain-bearing weather types, including Cyprus Lows and Red Sea Troughs. We further decomposed the

dynamic and thermodynamic contributions to changes in the water balance of precipitation minus evaporation, comparing the Last Interglacial peak with preindustrial time. Based on differences in daily mean precipitation, we provide evidence that the rain-bearing weather types yielded significantly more precipitation ($\approx +20\%$) during the Last Interglacial peak. This increase is most evident in the southern Levant, with higher precipitation during Red Sea Trough days, resulting primarily from thermodynamic changes. Minor differences in these weather types' characteristics were found. Our research provides insights into

historical hydroclimate changes in the Levant, extending our perspective on future climate impacts driven by natural variability.

## 1   Introduction

### 1.1   Overview of climate in the Levant

The hydroclimatic history of the Levant draws attention for two main reasons. First, it lies at the boundary between temperate northern and arid southern climates, rendering it highly responsive to small changes in the large-scale atmospheric circulation

[Enzel and Bar-Yosef (2017)]. Second, during the Last Interglacial peak, also known as Marine Isotope Stage 5e (MIS5e), the region has seen conditions that may have allowed humans to migrate "out of Africa." Increased rainfall transformed the



otherwise arid area into a viable passage, making it a critical migration gateway for early humans [Bar-Yosef (1998); Derricourt (2005); Schwarcz et al. (1988)].

The Levant is currently dominated by a Mediterranean-type climate with dry, hot summers and wet, cool winters [Goldreich
(2003); Armon et al. (2019)]. The precipitation season begins in October and ends in May, with most rainfall occurring during the winter months from December to March [Goldreich (2003)]. The average annual rainfall in the northern Levant and mountainous areas exceeds 1000 mm y$^{-1}$ with $\approx$ 70 rainy days per year [Goldreich (2003); Armon et al. (2019)]. In contrast, in the arid southeastern areas, rainfall can be well below 100 mm y$^{-1}$ [Vaks et al. (2007); Armon et al. (2019)].

The weather-types of the Levant have been classified into five main groups, including Cyprus Lows, abundant during winter;
Red Sea Troughs that peak in autumn; Persian Troughs exclusively occurring during summer; high-pressure systems throughout the year; and Sharav Lows in spring. Cyprus Lows and Red Sea Troughs are the main rain-bearing weather types [Fig. 1 A, B;Alpert et al. (2004b, a); Hochman et al. (2018b, a); Saaroni et al. (2010); Ziv et al. (2022)]. Cyprus Lows account for most annual precipitation (80 - 90%) and extreme weather events, while Red Sea Troughs contribute the rest [Goldreich (2003); Saaroni et al. (2010)]. Specifically, $\approx 50\%$ of precipitation in the hyper-arid areas of the Levant, such as Eilat, the southernmost
city in modern Israel, is due to the Red Sea Trough weather type [Saaroni et al. (2010); Ziv et al. (2022)]. In addition to these, the Tropical Plume is considered a rare rain-bearing sub-category that can transport large amounts of moisture from equatorial regions to the Levant [Ziv (2001); Rubin et al. (2007); Kahana et al. (2002)].

The northern Levant is projected to become warmer and dryer during the 21st century, associated with a projected decrease in the frequency of the Cyprus Low and Red Sea Trough and associated rainfall along with significant changes in their sea-
40 sonal occurrence [Hochman et al. (2018b, a, 2021); Armon et al. (2022)]. However, more southerly locations are projected to become wetter, with thermodynamic processes proposed as a plausible cause [Hochman et al. (2018c)]. Extreme precipitation events are projected to be shorter and more intense, with smaller rain areas and higher peak rain rates [Armon et al. (2022)].

## 1.2 Proxy-based hydro-climate reconstruction of the Last Interglacial

The Last Interglacial peak (approximately 127 ka) was characterized by elevated global average temperatures, higher sea levels, and increased atmospheric CO2 concentrations compared to glacial periods [Otto-Bliesner et al. (2021); Dutton and Lambeck (2012); Govin et al. (2015); Jouzel et al. (2007)]. During the Last Interglacial (LIG), the Levant experienced a relatively dry climate characterized by shorter, more intense rainfall events [Torfstein et al. (2015, 2013); Kushnir et al. (2024)]. Proxy-based reconstructions have indicated that the southern Levant experienced relatively wet conditions during this period, contrasting
with the generally arid climate prevalent throughout much of the Last Interglacial. Evidence supporting these wetter conditions comes from sediment cores obtained from the Dead Sea [Torfstein et al. (2013, 2015); Kiro et al. (2020); Kushnir et al. (2024)] and the Red Sea [Hartman et al. (2020); Palchan et al. (2018)], paleolakes in Jordan and Saudi Arabia [Petit-Maire et al. (2010); Rosenberg et al. (2013); Cordova et al. (2013)], speleothem records from caves [Vaks et al. (2003, 2006, 2007, 2010); Fleitmann et al. (2003, 2011); Burns et al. (1998); McGarry et al. (2004); Bar-Matthews (2014); Bar-Matthews et al. (2003)],



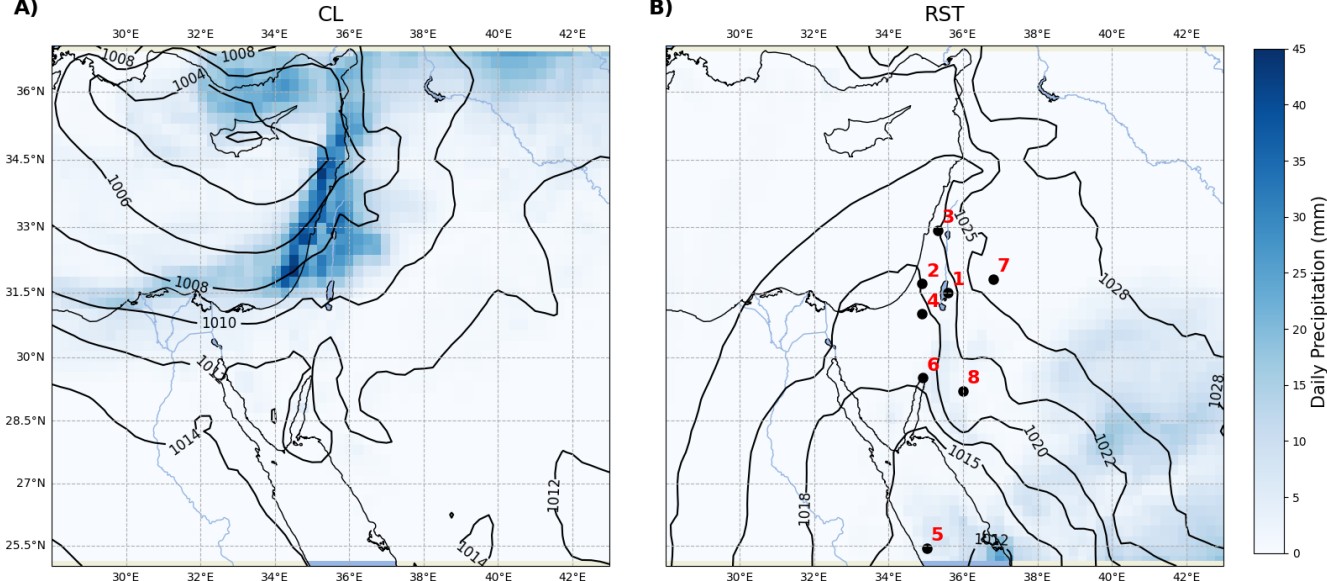

**Figure 1. Mean sea level pressure and precipitation maps of rain-bearing weather type examples in the Levant** A) Cyprus Low on 31 January 1992. B) Red Sea Trough on 10 January 1992. The location of geological archives for the Last Interglacial peak is in red: 1. Dead Sea [Torfstein et al. (2013, 2015); Kiro et al. (2020)], 2. Soreq Cave [Bar-Matthews et al. (2003); Bar-Matthews (2014)], 3. Pequin Cave [Bar-Matthews et al. (2003); Bar-Matthews (2014)], 4. In the square: Negev Cave deposits[Vaks et al. (2003, 2006, 2007, 2010)], 5. Red Sea core KL23 [Hartman et al. (2020); Palchan et al. (2018)], 6. Fossilized corals[Lazar and Stein (2011)],7. Azraq Oasis[Cordova et al. (2013)], 8. Mudawara Petit-Maire et al. (2010))

and coral studies[Lazar and Stein (2011)]. The presence of cave deposits in the southern Levant dating back to MIS 5e [Vaks et al. (2007, 2003, 2006)] suggests that their formation occurred under a hydrological regime significantly different from the present. Modern hydroclimatic conditions in the region do not support active speleothem growth, indicating that their development requires wetter and more favorable conditions than those currently prevailing.

    Additional insights come from the Dead Sea Deep Drilling Project (DSDDP), which retrieved a sediment core capturing
climatic conditions over the past 220,000 years [Torfstein et al. (2015); Kiro et al. (2020)]. This core has revealed alternating aragonite and detrital laminae during the peak of the Last Interglacial, indicative of a wetter climate driven by increased freshwater inflow [Torfstein et al. (2015)]. In contrast, halite layers observed before and after this peak reflect the dominance of hyper-arid conditions in the long term. Several proxy-based studies have proposed potential drivers for the changes in the hydroclimatic conditions observed at the Last Interglacial peak [Kiro et al. (2020); Palchan et al. (2018)]. These conditions
have been thought to have arisen from significant changes in the frequency of rain-bearing weather-types and a northward shift of tropical moisture sources toward the Levant [Kiro et al. (2020)]. For example, recently extracted sediment cores from the Red Sea have revealed increased local flooding events, enhanced soil transport, and elevated dust deposition from surrounding





regions over time. This trend has been attributed to a higher frequency of "wet" Red Sea Trough (RST) or Tropical Plume (TP) weather patterns [Palchan et al. (2018); Hartman et al. (2020)]. Another likely contributing factor is the increased occurrence of Mediterranean cyclonic systems, which could have triggered intense rainfall and widespread flooding across the region [Vaks et al. (2010); Kutzbach et al. (2020); Tierney et al. (2022)].

Analysis of the Dead Sea sediment core suggests an additional potential driver for the wetter conditions observed during the Last Interglacial peak. Located north of the Red Sea, current precipitation in the Dead Sea basin is primarily influenced by cyclonic systems originating from the Mediterranean [Goldreich (2003); Hartman et al. (2020)]. However, studies of the DSDDP core indicate a secondary moisture source originating from the tropics [Kiro et al. (2020); Torfstein et al. (2015)]. This tropical source is estimated to have contributed approximately 50% of precipitation during the Last Interglacial, likely due to a weakening of the Mediterranean influence and a concurrent strengthening of tropical moisture transport. This shift is attributed to the intensification of the African Summer Monsoon during the peak of the Last Interglacial [Kiro et al. (2020); Otto-Bliesner et al. (2021); Kutzbach et al. (2020); Orland et al. (2019)].

## 1.3 Model-based hydro-climate reconstruction of the Last Interglacial

Proxy records, which primarily capture coarse temporal and spatial scales in the late Quaternary and tend to be spatially limited, offer only a partial understanding of the mechanisms behind hydroclimatic variations in the Levant during the peak of the Last Interglacial [Ludwig and Hochman (2022); Otto-Bliesner et al. (2021); Kutzbach et al. (2020); Otto-Bliesner et al. (2013); Ludwig et al. (2019)]. To overcome these limitations, integrating proxy data with climate models provides a robust approach to exploring the drivers of paleoclimatic variability [Kiro et al. (2020); Kutzbach et al. (2020); Ludwig and Hochman (2022); Otto-Bliesner et al. (2021); Consortium et al. (2017); Fischer et al. (2018); Jones et al. (2009)]. Although relatively few studies have used this approach for the Levant, climate model simulations have provided significant insights [Ludwig and Hochman (2022)]. For example, increased precipitation in the Levant, particularly during summer, has been proposed based on an ensemble of PMIP4 models [Otto-Bliesner et al. (2021)]. This change has been further quantified using the CCSM3 model to reach 0.5 mm per day [Kiro et al. (2020)]. In summary, findings from geochemical proxies and paleoclimate models have indicated relatively wetter conditions and changes in the source of precipitation in the southern Levant during the Last Interglacial peak. However, the specific mechanisms driving these changes and associated precipitation characteristics still need to be better understood.

This study aims to characterize the underlying mechanisms of observed hydroclimatic changes in the Last Interglacial peak using PMIP4 models, a weather-type classification, and dynamic-thermodynamic decomposition. The manuscript is organized as follows: Description of available reanalysis data and PMIP4 models (Sect. 2.1). The weather type classification method (Sect. 2.2). Moisture balance, i.e., Precipitation minus Evaporation and decomposition into its dynamic and thermodynamic components (Sect. 2.3). The Results are organized in three sections. Sect. 3 evaluates the PMIP4 model's ability to characterize the hydroclimate during the Last Interglacial peak. Sect. 2.2 provides insights into weather type characteristics during the Last Interglacial peak, focusing on the dominant rain-bearing weather types. Sect. 3.3 portrays the changes in moisture balance and



the role of dynamics vs. thermodynamics in these. Sect. 5 provides the main findings and interpretation of the model results concerning proxy data.

## 2 Data and methods

### 2.1 Data

We used data from nine Global Circulation Models (GCM) from the 4th Phase of the Paleo-climate Model Inter-comparison Project [PMIP4; Kageyama et al. (2017)]. PMIP4 is an ongoing international research initiative studying past climates using model simulations. It aims to improve our understanding of Earth's climate system by simulating and comparing various climate models from different periods in Earth's history. Most of our analysis was based on the Alfred Wagner Institute Earth System Model [AWI-ESM; Sidorenko et al. (2019)] and the 3rd generation of the European Community Earth System Model

[EC-ESM; Hazeleger et al. (2010)] available in daily temporal resolution and horizontal grid spacing of 2.5° ($\approx 280$ km) and 1° ($\approx 111$ km), respectively. The analysis was based on 40-year model runs for each period. The models were chosen depending on data availability for the weather type classification (see Sect. 2.2). To assess our findings with respect to a larger ensemble of models in terms of moisture balance (see Sect. 2.3), we used all available PMIP4 models in monthly temporal resolution. The list of models is provided in Table 1. We evaluated the model's ability to capture the precipitation values

and weather type characteristics in the European Center for Medium-range Weather Forecast (ECMWF) reanalysis (ERA5) available at six hourly temporal resolutions and a horizontal grid-spacing of 0.25° ($\approx 31$ km) for 1981-2020 [Hersbach et al. (2020)]). A bootstrap test [Tibshirani and Efron (1993)] was employed to assess the significance of differences in hydroclimate variables between the Last Interglacial peak and Pre-Industrial periods, using a 5% significance level to evaluate precipitation differences.

| Model Name | Institution | Reference |
|---|---|---|
| AWI-ESM-1-1-LR | Alfred Wegener Institute (AWI), Germany | [Sidorenko et al. (2019); Shi et al. (2020)] |
| FGOALS-g3 | Chinese Academy of Sciences (CAS), China | [He et al. (2020); Zheng et al. (2020)] |
| ACCESS-ESM1-5 | Australian Community Climate and Earth System Simulator + University of New South Wales, Australia | [Ziehn et al. (2020); Yeung et al. (2021)] |
| MIROC-ES2L | Research Institute for Global Change, Japan | [Hajima et al. (2020); Ohgaito et al. (2020)] |
| NorESM1-F | Norwegian Climate Centre (NCC), Norway | [Guo et al. (2019)] |
| NorESM2-LM | Norwegian Climate Centre (NCC), Norway | [Seland et al. (2020); Zhang et al. (2019)] |
| CESM2 | National Center for Atmospheric Research (NCAR), USA | [Danabasoglu et al. (2020)] |
| NESM3 | Nanjing University of Information Science and Technology (NUIST), China | [Cao et al. (2018); Jian et al. (2019)] |
| EC-Earth3-LR | European Community(EC), Europe | [Hazeleger et al. (2010); Zhang et al. (2020b)] |

**Table 1.** List of nine PMIP4 models and their institutions and country.



## 2.2 Weather-type classification

We employed a semi-objective synoptic classification algorithm, identifying five primary weather types in the eastern Mediterranean: Persian Troughs, Highs, Sharav Lows, Red Sea Troughs, and Cyprus Lows. This classification successfully captures the hydroclimate conditions in the Levant [Alpert et al. (2004a); Hochman et al. (2018a)]. The classification uses four atmospheric variables at 1000hPa pressure level, which are geopotential height, temperature, zonal and meridional winds from the ERA5 reanalysis and two GCM simulations (AWI-ESM and EC-ESM; see Sect. 2.1) in the eastern Mediterranean (27.5–37.5 °N; 30–40 °E; Fig. 1). We compared the average Euclidean distances for the different periods to assess if the weather types during the Last Interglacial peak changed or were similar to today's. Our findings indicate that the average Euclidean distances between the Last Interglacial peak and preindustrial or ERA5 periods change by less than 5%, supporting the consistency of weather type characteristics across the different periods. The reader is referred to other studies for detailed information on the classification procedure [Alpert et al. (2004a, b); Hochman et al. (2018b, a); Ludwig and Hochman (2022)]. The binomial test was applied to compare the proportions of weather-type frequencies during the Last-Interglacial peak and Pre-Industrial periods at a 5% significance level.

## 2.3 Decomposing moisture balance into dynamic and thermodynamic components

The increase in wet conditions reflects a shift in water flux. The net water flux, defined as precipitation (P) minus evaporation (E) over a given surface, plays a fundamental role in the water cycle. While the globally averaged P minus E should theoretically balance to zero in current and future climates, assuming a closed system. However, regional variations can arise due to dynamic and thermodynamic processes influenced by climate change.

We employed a well-established method [Seager et al. (2010)] to analyze these variations and decompose the dynamic and thermodynamic components of moisture balance between the Last Interglacial peak and the Pre-Industrial period, using the AWI-ESM model. This model was selected based on its alignment with proxy data [see Sect. 3; Seager et al. (2019); Elbaum et al. (2022); Adam et al. (2023)].

In this framework, holding the humidity field constant isolates variations attributed to changes in the dynamic component, while modifications in the humidity field, with a fixed wind field, reflect adjustments in the thermodynamic component [Seager et al. (2019); Elbaum et al. (2022)].

The precipitation-minus-evaporation balance is mathematically defined as follows:

$$\mathbf{P} - \mathbf{E} = -\frac{1}{g\rho_w} \nabla \cdot \sum_{k=1}^{k} \mathbf{u}_k \mathbf{q}_k dp_k \tag{1}$$

Where **g** is the acceleration of gravity, $\rho_w$ is the density of water, **u** is the horizontal component of the wind, **q** is the specific humidity, **p** is the pressure, and $\kappa$ is the vertical layers of the model. All variables are monthly averages (over-bars in Equation 2).





In Equation 2, the first term represents the large-scale, long-term mean moisture transport, computed using the monthly averages of wind ($\overline{\mathbf{u}_k}$) and specific humidity ($\overline{q}_k$). This term reflects persistent atmospheric circulation patterns, such as trade winds and monsoons, crucial in regulating regional and global moisture distribution.

The second term captures short-term variability driven by transient eddies ($\mathbf{u}_k^{'}$ and $q_k^{'}$), which represent atmospheric disturbances such as storms and cyclones. These fluctuations are key in moisture transport, particularly within mid-latitude storm

tracks, contributing to episodic and intense precipitation events.

$$\overline{\mathbf{P}} - \overline{\mathbf{E}} = -\frac{1}{g\rho_w} \left[ \nabla \cdot \left( \sum_{k=1}^{k} \overline{\overline{\mathbf{u}_k \mathbf{q}_k} \overline{dp_k}} \right) + \nabla \cdot \left( \sum_{k=1}^{k} \overline{\mathbf{u}_k^{'} \mathbf{q}_k^{'} dp_k} \right) \right] \tag{2}$$

In practice, we focused on the change ($\Delta$) between the Last Interglacial peak and Pre-Industrial periods (Equation 3).

$$\Delta(\overline{P} - \overline{E}) \approx -\frac{1}{g\rho_w} \sum_{k=1}^{K} \Delta \overline{(\overline{\mathbf{u}}_k \cdot \nabla \overline{q}_k)} dp_k - \frac{1}{g\rho_w} \sum_{k=1}^{K} \Delta \overline{(\overline{q}_k \nabla \cdot \overline{\mathbf{u}}_k)} dp_k \tag{3}$$

The right-hand side of Equation 3 can be decomposed into three key components: dynamic, thermodynamic, and eddy

variability.

**Dynamic Component** – Represents changes in wind fields, capturing shifts in atmospheric circulation patterns that influence moisture transport (Equation 4).

**Thermodynamic Component** – Reflects variations in the moisture content driven by temperature and humidity changes, affecting the atmospheric water's overall holding capacity (Equation 5).

**Eddy Variability Component** – Derived from the second term in Equation 2, this component accounts for short-term fluctuations caused by transient eddies, such as storms and cyclones, which enhance episodic moisture transport (Equation 6).

$$\Delta\text{Dynamic} = -\frac{1}{g\rho_w} \sum_{k=1}^{K} \Delta \left( \overline{\mathbf{u}}_k \, dp_k \right) \cdot \nabla \overline{q}_{k,p_i} - \frac{1}{g\rho_w} \sum_{k=1}^{K} \overline{q}_{k,p_i} \Delta \left( \nabla \cdot \overline{\mathbf{u}}_k \, dp_k \right) \tag{4}$$

$$\Delta\text{Thermodynamic} = -\frac{1}{g\rho_w} \sum_{k=1}^{K} \overline{\mathbf{u}}_{k,p_i} \cdot \Delta \left( \nabla \overline{q}_k \, dp_k \right) - \frac{1}{g\rho_w} \sum_{k=1}^{K} \nabla \cdot \overline{\mathbf{u}}_{k,p_i} \Delta \left( \overline{q}_k \, dp_k \right) \tag{5}$$

$$\Delta\text{Eddys} = -\frac{1}{g\rho_w} \sum_{k=1}^{K} \Delta \left( \nabla \cdot \left( \mathbf{u}_k^{'} q_k^{'} \, dp_k \right) \right) \tag{6}$$





## 3 Results

### 3.1 Evaluating PMIP4 models concerning proxies and reanalysis

First, we evaluated the PMIP4 models to assess their reliability compared to proxy-based reconstructions (see Sect. 1.3). Previous studies comparing precipitation between AWI-ESM and ERA5 reanalysis indicated that AWI-ESM effectively captures the Levant's hydro-climate [Ludwig and Hochman (2022)]. The two climate models diverge in representing seasonal precipitation differences between the Last Interglacial peak and Pre-Industrial periods (Fig. 2). AWI-ESM suggests wetter winters in the Levant basin compared to the Pre-Industrial period, consistent with speleothem evidence from the Negev [Vaks et al. (2007)]. In contrast, EC-ESM shows wetter winters confined to the northern Levant. During summer, the models differ significantly: AWI-ESM indicates increased precipitation, particularly in the southern Levant, aligned with evidence of intensified weathering near the Red Sea [Palchan et al. (2018)], and transitions in Dead Sea sedimentation from salt to laminated silts[Kiro et al. (2020); Torfstein et al. (2015)]. However, EC-ESM shows a further southward change in summer precipitation. Autumn patterns also vary: AWI-ESM indicates drying in the northern Levant and increased precipitation in the south, while EC-ESM suggests widespread precipitation increases across the northern region. In spring, both models show a slight increase in precipitation; the AWI-ESM model indicates an increase mainly in the northern Levant, while the EC-ESM model shows a more pronounced increase primarily in the southern Levant. (Fig. 2).

The models exhibit distinct precipitation patterns. In AWI-ESM, autumn and summer precipitation is primarily concentrated in the southern regions (Fig. 2 A, D), aligning with evidence of increased rainfall intensity and enhanced weathering in the Red Sea and Dead Sea areas, driven by tropical systems [Palchan et al. (2018); Kiro et al. (2020)]. In contrast, EC-ESM depicts minimal changes, with precipitation shifts occurring mainly in the northern part of the domain. (Fig. 2 E, H.).

We evaluated weather-type frequencies to explore potential drivers of precipitation differences between periods. Both models effectively capture seasonal-scale weather-type frequencies compared to ERA5 (Fig. 3 A-H). In AWI-ESM, the frequency of Cyprus Lows increases during winter, occurring on about 50% of winter days. At the same time, no significant changes are observed in other weather- types during autumn or spring (Fig. 3 B). Summer shows no changes in known rain-bearing systems, but the Persian Trough increases its occurrence, affecting over 90% of days during the Last Interglacial peak. In EC-ESM, Red Sea Trough frequencies increase in autumn, but neither model exhibits statistically significant weather-type frequency changes between the Last Interglacial peak and Pre-Industrial periods.

Determining the factors driving model differences is complex, but several key explanations emerge. AWI-ESM incorporates interactive vegetation, whereas EC-ESM relies on prescribed vegetation modules. Additionally, EC-ESM exhibits reduced winter ice extent during the Last Interglacial peak compared to the Pre-Industrial period [Kageyama et al. (2021)], contributing to a larger positive temperature anomaly relative to other PMIP4 models, including AWI-ESM [Otto-Bliesner et al. (2021)]. Given that EC-ESM diverges from other models and proxy reconstructions [Zhang et al. (2020a)], we prioritized AWI-ESM, which more reliably captures the Levant's hydroclimate during both the Last Interglacial and Last Glacial Maximum peaks [Ludwig and Hochman (2022)].





**Figure 2.** Last Interglacial - Pre-Industrial precipitation [mm/d]. For every season: Autumn- September, October, November [SON - A, E]. Winter- December, January, February [DJF - B, F]. Spring- March, April, May [MAM - C, G]. Summer- June, July, August [JJA - D, H]. In two models, AWI-ESM [A-D] and EC-ESM [E-H]. Colored regions denote statistically significant changes at the 5% level following a bootstrap test.





## 3.2 The hydroclimate during the Last Interglacial peak

Analysis of precipitation by weather type suggests a 17.3% increase in the daily average precipitation during Cyprus Low days
in the AWI-ESM model compared to the Pre-Industrial period, with the most pronounced increase observed in the northern
Levant, particularly over Turkey (Fig. 4A). Similarly, a quantitative analysis shows a 23.3% increase in daily mean precipitation
during Red Sea Trough days, derived from a direct comparison of mean daily precipitation between the two periods. While the
figures do not explicitly display these computed values, Figure 4B highlights the spatial distribution of precipitation changes,
illustrating areas where the daily average precipitation during Red Sea Trough days increased, particularly around the Red Sea,
but also in other parts of the Levant. In contrast, during Cyprus Low days, the increase in precipitation is mainly confined to
the northern Levant. This increase is most pronounced in autumn, winter, and spring, while summer sees a minor impact due
to the relatively low frequency of Red Sea Trough occurrences during this season (Fig. 3).

High-percentile precipitation events (90th percentile of precipitation days) exhibit even greater differences between the
Last Interglacial peak and the Pre-Industrial period, with significantly higher precipitation intensities recorded during Cyprus
Low (Fig. 4C) and Red Sea Trough (Fig. 4D) days. Notably, this increase is most evident in the northern Levant, where the
frequency and intensity of high-percentile precipitation events have risen. In contrast, while localized increases are observed in
the southern Levant, high-percentile precipitation values have generally remained unchanged or even declined in some areas.

This analysis was conducted annually, considering all Cyprus Low or Red Sea Trough days. As shown in Fig. 3, these
systems are predominantly active during autumn, winter, and spring, with a limited presence in summer.

At first glance, Figure 4 contrasts proxy-based findings, indicating a general increase in average and High-percentile pre-
cipitation events across the Mediterranean region. However, a closer examination reveals a more nuanced pattern. The data
show that this increase is primarily associated with Cyprus Lows, a characteristic Mediterranean weather system. Additionally,
precipitation linked to Red Sea Troughs increases, but these systems can draw moisture from either the Mediterranean, like
Cyprus Lows, or southern moisture sources [Tsvieli and Zangvil (2007); Hochman et al. (2023)]. Proxy-based studies have
frequently highlighted increased precipitation from southern sources rather than Mediterranean ones. This suggests that the
observed precipitation characteristics on Red Sea Trough days may reflect a shift in moisture sources, aligning with proxy
evidence of enhanced contributions from the south.

To further characterize precipitation variability during the Last Interglacial peak, we examine transitions between weather-
types using the weather type transition probability matrix (Fig. 5). This matrix quantifies the likelihood of one weather type
transitioning to another the following day, allowing a comparison between the Last Interglacial peak and the preindustrial
period across different seasons and on an annual scale.

Our analysis reveals that Cyprus Lows exhibited 6.4% greater persistence in winter during the Last Interglacial peak (Fig.
5B) and a 4.8% increase annually (Fig. 5E). Meanwhile, Red Sea Troughs showed the largest increase in persistence during
autumn at 2.2% (Fig. 5A), although the annual change was minimal at 0.2%.




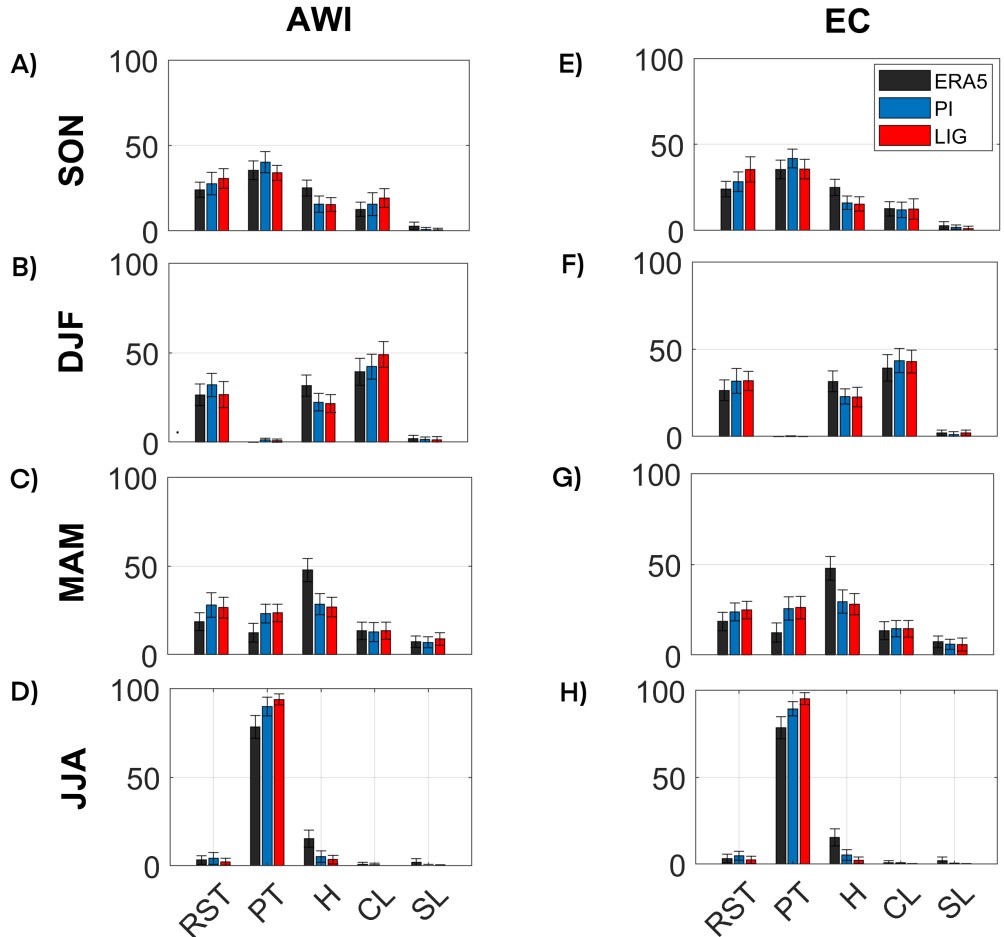

**Figure 3.** The seasonal frequencies of weather types in the Levant. Autumn- September, October, November [SON - A, E]. Winter - December, January, February [DJF - B, F]. Spring - March, April, May [MAM - C, G]. Summer - June, July, August [JJA - D, H]. For three time periods: ERA5 reanalysis [ERA5], Last Interglacial peak [LIG], and Pre-Industrial [PI]. The weather types are Red Sea Trough [RST], Persian Trough [PT], High-pressure systems [H], Cyprus Low [CL] and Sharav Low [S]. Two models were considered: the AWI-ESM [A-D] and the EC-ESM [E-H].

### 3.3 Identifying the drivers of hydro-climate changes in the Last Interglacial peak

Analyzing the moisture balance - the difference between precipitation and evaporation ($\Delta P - E$)—provides a deeper insight into the hydro-climatic differences between the Last Interglacial peak and the Pre-Industrial period. During winter and spring,

the moisture balance remains largely unchanged across most of the Levant, with some localized decreases, particularly over modern Israel. In contrast, summer exhibits a notable increase, suggesting higher water availability in the southern Levant




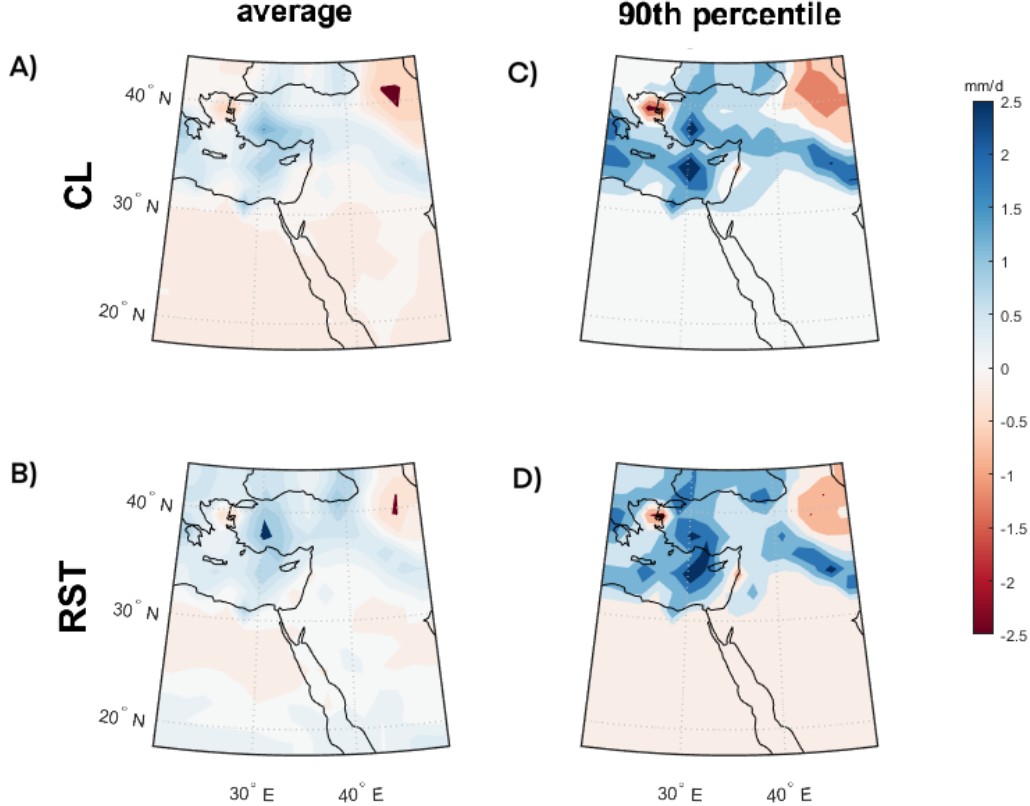

**Figure 4.** Differences in average and 90th percentile of precipitation [mm/d] for Red Sea Trough [RST] and Cyprus Low [CL] between the Last Interglacial peak [LIG] and Pre-Industrial [PI] periods. Colored regions denote statistically significant changes at the 5% level following a bootstrap test.

and the northern Red Sea. This trend extends into autumn, with a continued, though less pronounced, increase over Northeast Africa (modern Egypt; Fig. 6).

   When decomposing these changes into dynamic and thermodynamic components, we find that the summer increase in moisture balance is primarily driven by the thermodynamic component, indicating higher moisture availability in the Levant (Fig. 7 [J - L] ). This increase is consistent with previous studies [Otto-Bliesner et al. (2021)] on the Last Interglacial peak,
highlighting that most of the warming occurred during summer. Across different models, summer temperatures in the Levant rose by more than 4°C to 7°C. In contrast, winter temperatures exhibited a cooling trend [Otto-Bliesner et al. (2021)], likely contributing to a reduction in the moisture balance (Fig. 7 D - F). This seasonal contrast emphasizes the critical role of the thermodynamic component in summer, as warmer air can hold more water vapor, thereby enhancing its ability to transport moisture into the Levant.





| AWI SON LIG-PI | | | | | |
|---|---|---|---|---|---|
| | probability for the next day | | | | |
| system | RST | PT | H | CL | SL |
| RST | 2.2 | -2.9 | -2.0 | 3.5 | -0.7 |
| PT | -0.4 | -0.7 | 0.3 | 0.5 | 0.2 |
| today H | 0.3 | -2.4 | 0.4 | 2.2 | -0.7 |
| CL | 3.2 | -3.0 | -1.6 | 1.5 | -0.1 |
| SL | -8.3 | 5.2 | 13.5 | -17.7 | 7.3 |

| AWI MAM LIG-PI | | | | | |
|---|---|---|---|---|---|
| | probability for the next day | | | | |
| system | RST | PT | H | CL | SL |
| RST | 1.6 | -4.2 | 2.1 | 0.4 | -0.1 |
| PT | -2.6 | 5.0 | -5.4 | 0.6 | 2.3 |
| today H | -1.3 | 0.9 | -2.5 | 1.3 | 1.7 |
| CL | -1.4 | -1.6 | -0.2 | 1.8 | 1.5 |
| SL | -3.1 | 1.9 | -1.0 | -1.4 | 3.9 |

| AWI DJF LIG-PI | | | | | |
|---|---|---|---|---|---|
| | probability for the next day | | | | |
| system | RST | PT | H | CL | SL |
| RST | -3.8 | -0.3 | -0.2 | 4.2 | 0.1 |
| PT | -8.3 | 0.0 | -15.1 | 28.1 | -4.1 |
| today H | -4.0 | -0.3 | 2.1 | 1.5 | 0.8 |
| CL | -4.5 | -0.3 | -0.5 | 6.4 | -1.1 |
| SL | -0.1 | -1.1 | -6.9 | 4.2 | 4.0 |

| AWI JJA LIG-PI | | | | | |
|---|---|---|---|---|---|
| | probability for the next day | | | | |
| system | RST | PT | H | CL | SL |
| RST | -17.3 | 22.4 | -4.4 | 0.0 | -0.7 |
| PT | -1.0 | 1.9 | -0.8 | -0.2 | 0.0 |
| today H | -0.1 | 6.6 | -4.4 | -1.0 | -0.8 |
| CL | 0.0 | 50.0 | -11.1 | -38.9 | 0.0 |
| SL | 16.7 | 45.8 | -62.5 | 0.0 | 0.0 |

| LIG-PI | | | | | |
|---|---|---|---|---|---|
| | probability for the next day | | | | |
| system | RST | PT | H | CL | SL |
| RST | 0.2 | -2.1 | -0.3 | 2.4 | -0.2 |
| PT | -1.4 | 2.6 | -1.4 | -0.1 | 0.4 |
| today H | -1.3 | -1.3 | -0.3 | 2.2 | 0.7 |
| CL | -2.0 | -1.4 | -1.0 | 4.8 | -0.4 |
| SL | -2.3 | 1.8 | -2.5 | -2.4 | 5.4 |

**Figure 5.** Weather type transition probability matrix. The difference between the Last Interglacial peak [LIG] and Pre-Industrial [PI] periods. The weather types are Red Sea Trough [RST], Persian Trough [PT], Highs [H], Cyprus Lows [CL], and Sharav Lows [SL]. For every season: Autumn [SON - A], Winter [DJF - B], Spring [MAM - C], Summer [JJA - D], and annual [E]. Statistically significant differences are marked with an underline using a binomial test at the 5% significance level.

The dynamic component remains largely unchanged and shows a decline in some regions, suggesting a weakening of eddy-driven contributions. This decline indicates that large-scale circulation patterns played a less significant role in moisture transport during this period. The reduced influence of the dynamic component is particularly pronounced in summer and autumn, reinforcing the notion that thermodynamic processes were the dominant drivers of moisture balance during these seasons.

In autumn and winter, the dynamic component contributes less to the overall moisture balance. However, in spring, it exhibits
a marked increase in the Levant, accounting for much of the seasonal rise in moisture availability (Fig. 7 B, E.). The position and strength of the jet stream were also analyzed, revealing no significant changes (not shown), further supporting the limited role of large-scale circulation shifts in driving variations in moisture balance.

These seasonal variations emphasize the complex interplay of atmospheric processes, highlighting a shifting influence between thermodynamic and dynamic factors throughout the year. The results are based on the multi-model mean of nine PMIP4
simulations, ensuring robustness and consistency across different climate models (see Table 1).





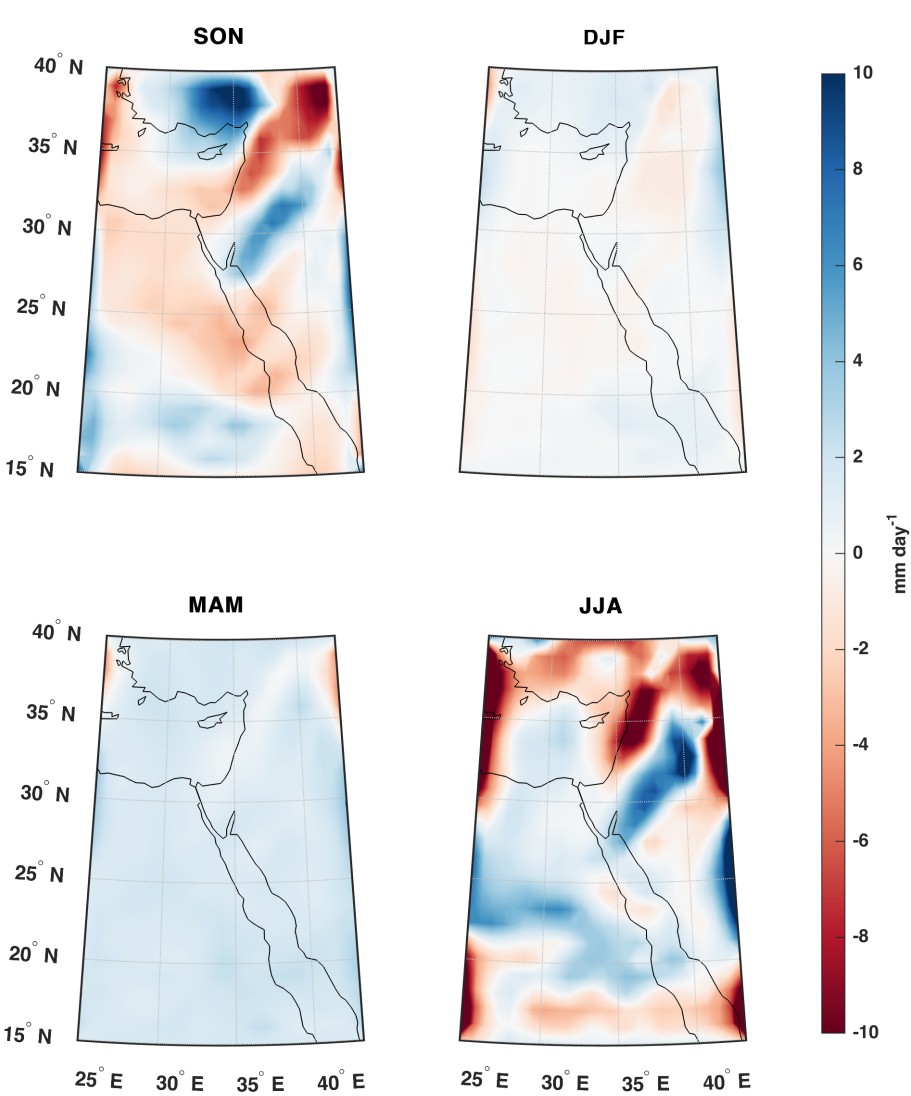

**Figure 6.** Delta Precipitation - Evaporation $\Delta P - E$ [mm/d] for the Last Interglacial peak [LIG] compared to the pre-industrial [PI] period. For every season: Autumn- [SON - A], Winter [DJF - B], Spring [MAM - C] and Summer [JJA - D]. Colored regions denote statistically significant changes at the 5% level following a bootstrap test.

## 4 Discussion

Proxy records indicate that while the Last Interglacial peak (MIS5e) was generally hyper-arid, it was punctuated by moisture intrusions from the south. Some studies link these shifts to a northward expansion of the African Summer Monsoon [Kiro et al.





(2020)]. However, we suggest that Red Sea Trough intensification better explains the observed changes, as it primarily affects the Levant in autumn and winter rather than summer. Thus, increased northern Levant precipitation likely reflects enhanced autumn and winter rainfall rather than wetter summers.

In contrast, regions near the Red Sea are influenced by the African Summer Monsoon and Tropical Plumes, which drive summer rainfall in East Africa and increase Nile River flow [Bar-Matthews et al. (2000); Rohling et al. (2002)]. The Red Sea Trough, however, remains active during transitional seasons and winter [Tsvieli and Zangvil (2005, 2007); Ziv et al. (2022)]. Both systems transport moisture from the south, but the Red Sea Trough exerts a broader climatic impact over the Levant. Indeed, there is a 23% increase in average precipitation during Red Sea Trough days. This increased southern precipitation during MIS5e likely resulted from intensified Red Sea Trough events, contributing to a more substantial climatic effect across the region.

Proxy records from central Israel do not strongly support greater summer rainfall but instead suggest higher precipitation intensity, likely due to a stronger Red Sea Trough, active from October to May. Climate models confirm this pattern, showing an increase in daily rainfall linked to the Red Sea Trough (Fig. 4), while summer rainfall increases remained localized along the Red Sea coast [Otto-Bliesner et al. (2021); Kutzbach et al. (2020)].

## 5 Summary and Conclusions

This study investigated the hydroclimate conditions in the Levant during the Last Interglacial peak, focusing on weather-type characteristics and the drivers of moisture balance. Using proxy-based paleoclimate reconstructions and climate models from the 4th phase of the Paleo-climate Model Inter-comparison Project (PMIP4), we analyzed seasonal moisture balance patterns, particularly in the southern Levant. Our results reveal that thermodynamic changes influenced moisture availability during the Last Interglacial peak, increasing atmospheric moisture capacity, especially in summer and autumn. Decomposing the moisture balance into thermodynamic and dynamic components shows that the thermodynamic contribution was the dominant factor in shaping seasonal moisture conditions. In contrast, changes in the dynamic component were minimal. The study also highlights the role of synoptic-scale weather-types, such as Cyprus Lows and Red Sea Troughs, in shaping regional moisture patterns. While an increase in moisture availability was observed during the Last Interglacial peak, only small changes were detected in the frequency of these weather-types. The study emphasizes the importance of integrating climate models with proxy data to improve understanding of past climate variability and its relevance for future climate projections. Despite advancements, we acknowledge limitations in model resolution and temporal coverage, suggesting that further investigations are necessary to refine our knowledge of regional hydroclimate processes. Ultimately, this study provides valuable insights into the dynamics of moisture balance during the Last-Interglacial and aids in developing future climate scenarios, considering both natural variability and anthropogenic influences.



*Data availability.* All data used in this study is publicly available. We acknowledge the climate modeling groups for producing and providing access to their model output, as well as the Earth System Grid Federation (ESGF) for archiving and facilitating data access. The results and conclusions presented here are based on two key datasets: the 5th generation of the European Centre for Medium-Range Weather Forecasts (ECMWF) ERA5 reanalysis dataset www.ecmwf.int/en/forecasts/datasets/reanalysis-datasets/era5 and multiple simulations from the Paleoclimate Model Intercomparison Project Phase 4 (PMIP4). We thank the multiple funding agencies that support PMIP and ESGF. All PMIP4 data analyzed in this study are available through the ESGF at https:https://esgf-node.llnl.gov/projects/esgf-llnl

*Author contributions.* The following is a detailed breakdown of each author's contributions to the manuscript. **EB** : Conceptualization; Methodology; Formal Analysis; Software; Validation; Data Curation; Writing – Original Draft; Visualization. **AT**: Supervision; Conceptualization; Methodology; Writing – Review & Editing; Funding Acquisition. **AH** : Supervision; Conceptualization; Methodology; writing – Review & Editing; Funding Acquisition. **RY**: Software; Validation.

*Competing interests.* The authors have no relevant financial or non-financial interests to disclose.

*Acknowledgements.* The Israel Science Foundation (grant 978/23), the Pazy Foundation (grant 434), the Federal Ministry of Education and Research (BMBF), Germany and the Ministry of Innovation Science and Technology of Israel within the GRaCCE project, the COST Actions CA19109 and CA22162, 'MedCyclones' and 'FutureMed' supported by COST (European Cooperation in Science and Technology), and the Nuclear Research Center of the Negev provide support for the contribution of AH and EB. The authors acknowledge using Grammarly (https://www.grammarly.com) for English editing purposes.



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





**Figure 7.** Eddy Dynamic and Thermodynamic decomposition of the difference between Precipitation- Evaporation $\Delta P - E$ [mm/d] in the Last Interglacial peak [LIG] compared to the Pre-Industrial [PI] period. For every season: Autumn [SON, A - C], Winter [DJF, D - F], Spring [MAM, G - I], and Summer [JJA, J - L]. Multi-model ensemble mean of the nine PMIP4 models. Colored regions denote statistically significant changes at the 5% level following a bootstrap test.