# Peer review of "Hydroclimatic Variability and Weather-Type Characteristics in the Levant During the Last Interglacial"

_EGUsphere, 2025_

## Referee Comment (RC2)

This manuscript investigates hydroclimatic variability in the Levant during the Last Interglacial peak (MIS 5e) using PMIP4 paleoclimate model simulations in combination with a synoptic weather-type classification and moisture balance decomposition. The study mostly uses the AWI-ESM model to examine precipitation characteristics and the relative roles of thermodynamic and dynamic processes. Results indicate enhanced precipitation—particularly during Red Sea Trough and Cyprus Low events—driven largely by thermodynamic factors. The authors present integration of model and proxy data but the manuscript requires major revisions to address issues of clarity, internal consistency, and the robustness of methodological and interpretative claims. The manuscript is interesting and deseves publication in Climate of the Past, provided the authors address the comments and improve its quality.

**Comments**

Line 47-50:

First claim:

"The Levant experienced a relatively dry climate characterized by shorter, more intense rainfall events..."

This statement describes the entire Levant as relatively dry.

Second claim:

"Proxy-based reconstructions have indicated that the southern Levant experienced relatively wet conditions during this period..."
This says that the southern Levant was relatively wet.

**Conflict:**

These two statements describe opposite hydroclimatic conditions (dry vs. wet) for the same region and period, unless the author means to emphasize spatial variability within the Levant (north vs. south, for instance).

So unless the text explicitly clarifies that the southern Levant differed from the rest of the Levant, the passage reads as internally inconsistent — the same time period is characterized both as relatively dry overall and relatively wet locally.

To fix it, you could clarify the regional contrast explicitly, for example:

"While the Levant as a whole experienced relatively dry conditions, proxy records suggest that the southern Levant may have been comparatively wetter during this period."

Line 111: 'The analysis was based on 40-year model runs for each period.' Which period, please mention.

In Table 1, please mention spatial resolution of each model. Please also mention for what time period the simulations are available. In Table 1 you listed 9 models, and then you mostly use only two models for your analysis, what is the justification for this?

Line 122: 'Sharav Lows', is it Sahara Lows.

Lines: 126-127: 'We compared the average Euclidean distances for the different periods to assess if the weather types during the Last Interglacial peak changed or were similar to today's.' what do you mean here by different pereiods, please mention the periods to make it clear. What length of interglacial period you have compared with ERA5, and preindustrial.

Line 128: What preindustrial period you have chosen, please mention. Line 140: the authors mention here the proxy data, the proxy data should be discussed in the data section before mentioning here. It is not clear which proxy is being used, and for what time period.

Fig. 2: Explain how you have appllied the bootstrap test in methods section. In sub-figure titles write the complete model name not just AWI, or EC, othewise mention these abrevations in the main text before using in the figures. In method section for clarity to the reader, please justify why you subtract interglacial period from the preindsutrial. If possible please demarkate Levant basin in Fig. 2.

Line 172: 'First, we evaluated the PMIP4 models to assess their reliability compared to proxy-based reconstructions (see Sect. 1.3).' I do not see any evaluation of PMIP4 models in Sect. 1.3.

Lines 175-177: 'AWI-ESM suggests wetter winters in the Levant basin compared to the Pre-Industrial period, consistent with speleothem evidence from the Negev [Vaks et al. (2007)]. In contrast, EC-ESM shows wetter winters confined to the northern Levant.' This statatemt is not consistent with Fig. 2b (winter) as I see similar wetter condtions for both models (AWI, and EC).

Lines 180-182: 'Autumn

patterns also vary: AWI-ESM indicates drying in the northern Levant and increased precipitation in the south, while EC-ESM suggests widespread precipitation increases across the northern region.' I do not find this statement consistent with Fig. 2A and 2E. Better demark Levant in the figure. It is not clear what you consider the Levant region, mention clearly in methods section.

Fig. 3: What the y-axis shows? In the main text, please, elaborate the sub-figure that comes first, not just randomly. Not clear what periods you have chosen for ERA5, preindsutrial, and interglacial, mention them also in the main text in methods section.

Line 189: 'We evaluated weather-type frequencies to explore potential drivers of precipitation differences between periods'. Which periods, not clear?

Line 190-192: 'In AWI-ESM, the frequency of Cyprus Lows increases during winter, occurring on about 50% of winter days. At the same time, no significant changes are observed in other weather- types during autumn or spring (Fig. 3 B).' What do you mean by changes/inrease here, is it the change in frequency, if so, then compared

to what, it is very confusing. While refering to results from the figures, please mention the sub-figure letter (e.g., Fig. 3A etc.) in the main text.

Lines: 204-206: 'Analysis of precipitation by weather type suggests a 17.3% increase in the daily average precipitation during Cyprus Low days in the AWI-ESM model compared to the Pre-Industrial period, with the most pronounced increase observed in the northern Levant, particularly over Turkey (Fig. 4A).' Isnt there also significant increase over parts of Mediterreanean sea.

Line 220: 'At first glance, Figure 4 contrasts proxy-based findings,'. Which proxy based findings, it is not clear.

Lines 224-224: 'Proxy-based studies have

frequently highlighted increased precipitation from southern sources rather than Mediterranean ones.' Which proxy based studies highlighted this, please cite them here.

Fig. 5 should be a table, not a figure. Further, the sub-figure numbers (e.g., 5A, 5C etc) are not marked.

In section 3.3 you are describing results with reference to the figure. please refer to the figure in the very first lines (236-239).

Lines 237-239: 'During winter and spring,

the moisture balance remains largely unchanged across most of the Levant, with some localized decreases, particularly over

modern Israel.' I this statement realy consistent with the Fig. 6 (DJF, MAM) when you say that colored regions show sinificant changes at 5% level, thus what I understand from this the changes, even thoug small, are still significant.

Line 280: 'Using proxy-based paleoclimate reconstructions and climate models ...' I do not see usage of proxy-based paleoclimate reconstructions in this manuscript, please clarify, how you used it.

---

## Author Comment (AC2)

**Point-by-point response**

**'Comment on egusphere-2025-3088'**

**Reviewer 1:** This study by Bril et al. presents an in-depth analysis of hydroclimate variability and weather regimes in the Levant during the Last Interglacial. The authors use PMIP4 model simulations under Last Interglacial forcing and compare them to pre-industrial conditions representing the current interglacial without anthropogenic influence. Their results indicate distinctly wetter conditions during the Last Interglacial, driven by an increase of approximately 20% in precipitation associated with rain-bearing weather regimes due to thermodynamic changes.

The methods are generally appropriate, the study region and time period are of high scientific relevance (e.g., implications for human migration during MIS5e), and the conclusions are supported by strong modelling evidence. The use of weather-regime clustering to differentiate hydroclimatic drivers is particularly valuable and insightful. The eddy/thermodynamic/dynamic decomposition is also an elegant and insightful analysis. However, the experimental setup and data/methods descriptions are at times unclear, making it difficult for the reader to fully follow the workflow. With revisions to improve clarity, this study is suitable for publication in Climate of the Past. Below, I outline two (moderate) major comments and several minor comments.

**Response:** We would like to thank the reviewer for the time and effort invested in reviewing our manuscript and for the insightful comments and suggestions. We are pleased that the reviewer found our study interesting. The comments have been very helpful in improving the clarity and overall quality of the manuscript.

During the review, the reviewer's comments will be in black, our response will be in red, and the revised text will be in italic point by point.

**Major Comments**

1. Clustering methodology

The clustering approach-referred to briefly as a "semi-synoptic classification algorithm" is insufficiently described. It is not clear whether this is a specific established method or an adaptation of previous approaches. Given that this clustering strategy is central to the study and is not widely used (to my knowledge), a more detailed methods description is necessary. The authors should explain how the clustering is performed, provide references for the algorithm, and justify its selection over more commonly used approaches (e.g., k-means, DBSCAN, self-organizing maps).

**Response:** The synoptic classification method was originally developed by Alpert et al. (2004) and has since become a standard approach for identifying and classifying synoptic systems in the Eastern Mediterranean. It has been widely applied and refined in numerous regional studies. In the revised manuscript, we added more technical details describing how this method was implemented in our analysis. As noted in the paper, we chose to rely on this well-established classification scheme rather than develop a new one, ensuring consistency with previous work and reliable representation of the region's seasonality and synoptic features. The revised text follows L125-140:

"We employed the semi-objective synoptic classification algorithm originally developed by [Alpert et al. (2004b) which has been widely used in studies of the eastern Mediterranean climate [Alpert et al. (2004a, b); Hochman et al. (2018b, a); Ludwig and Hochman (2022)]. The method is based on a reference archive of 426 days (1985 and winter 1991–1992), which five expert forecasters subjectively classified into 19 synoptic types. These types were later grouped into five primary weather types (daily scale): Persian Troughs, Highs, Sharav Lows, Red Sea Troughs, and Cyprus Lows. For each day, a feature vector is constructed from four near-surface atmospheric variables, geopotential height, temperature, and the zonal and meridional wind components, at 1000 hPa, averaged over a 5×5 grid covering the eastern Mediterranean domain (27.5–37.5° N, 30–40° E; Fig. 1). Days from ERA5 and from the GCM simulations (AWI-ESM and EC-ESM; see

Sect. 2.1) were then classified by assigning each to the expert-labeled reference day with the minimum Euclidean distance in this multidimensional space. This approach is considered semi-objective because it combines automated distance-based classification with limited expert verification, ensuring physical consistency of the resulting patterns. It has been shown to reproduce regional hydroclimatic variability with high fidelity. It provides physically interpretable weather types, in contrast to unsupervised clustering methods that may produce clusters lacking clear synoptic meaning. We compared the average Euclidean distances among the three reference periods—the Last Interglacial peak, the Pre-Industrial period, and the ERA5 reanalysis—to assess whether the weather-type characteristics changed. The binomial test was then applied to compare the proportions of weather-type frequencies between the Last Interglacial and Pre-Industrial periods at the 95% significance level."

**2. Model selection and usage**

The current presentation of the climate models is confusing: multiple models are listed and none of them is mentioned in the text. Meanwhile, the text emphasizes the use of only two models, and the analysis in section 2.3 appears to use only AWI-ESM. It is unclear whether this choice is motivated by model performance, data availability, or other practical considerations. Together, it is hard to tell when and where the authors use which model. Additionally, Table 1 lists models that are not discussed in the main text. The authors should clearly state which models are used for which analyses, why certain models are prioritized, and how these decisions affect interpretation.

**Response 2**: Thank you for this important comment. In this study, we examined hydroclimatic variability from several perspectives:

1. Weather-type characteristics: analyzed using models with daily resolution for classification purposes. For this part, we used two models (AWI-ESM and EC-Earth). After observing that the AWI-ESM model produced results consistent with

- the proxy records, we focused on this model to further investigate which weather type could explain the proxy-inferred patterns namely, the enhanced southern contribution and higher rainfall intensities as discussed in Section 3.2 of the Results.
- 2. Moisture-balance analysis following the identification of a potential southern moisture contribution in the AWI-ESM analysis, we sought to determine whether the observed changes were driven dynamically or thermodynamically. This required analyzing the moisture balance, which could be performed using monthly rather than daily data. The lower temporal resolution enabled us to include a larger ensemble of models, allowing a more robust decomposition of the precipitation—evaporation balance. Accordingly, instead of focusing solely on two models with an in-depth analysis of one, we extended the investigation to additional PMIP4 models in the monthly-scale analysis, as described in Section 3.3 of the Results. Table 1 lists the models used in this section.

To improve clarity, we added several clarifying passages to the Data section, which now reads as follows, L105-124:

"We used data from nine General Circulation Models (GCM) contributing to the 4th Phase of the Paleoclimate Model Inter-comparison Project [PMIP4; Kageyama et al. (2017)]. PMIP4 is an ongoing international research initiative studying past climates using model simulations. It aims to improve our understanding of Earth's climate system by simulating and comparing various climate models from different periods in Earth's history. Most of our analysis was based on the Alfred Wagner Institute Earth System Model [AWI-ESM; Sidorenko et al., 2019] and the 3rd generation of the European Community Earth System Model [EC-ESM; Hazeleger et al., 2010] available in daily temporal resolution and horizontal grid spacing of 2.5° (≈ 280 km) and 1° (≈ 111 km), respectively. The analysis was based on 40-year model runs for each period. Daily-resolution data were available only for the AWI-ESM and EC-Earth models, which

were therefore used for the weather-type classification described in Section 2.2. All available PMIP4 models were used at monthly resolution to evaluate the large-scale moisture balance (see Sect. 2.3), as this temporal resolution was adequate for this purpose. The list of models is provided in Table 1. For weather type analysis, Sect. 2.2, we evaluated each model's ability to capture the precipitation values and weather type characteristics

in the European Center for Medium-range Weather Forecast (ECMWF) reanalysis (ERA5) available at six hourly temporal resolutions and a horizontal grid-spacing of 0.25° (≈31 km) for 1981-2020 [Hersbach et al., 2020]. A bootstrap test [Tibshirani and Efron, 1993]. To assess the statistical significance of differences in precipitation between the Last Interglacial (LIG) and Pre-Industrial (PI) periods, we applied a non-parametric bootstrap resampling test [Tibshirani and Efron, 1993]. In this approach, repeated random resampling with replacement was performed 1000 times within each period to generate a new empirical distribution of mean precipitation differences based on the original data. Statistical significance was determined at the 95% level, with differences considered significant when zero was outside the 95% confidence interval."

In addition, we made corrections throughout the Results and Discussion sections to clarify which model is being referred to in each case, as suggested in the minor comment.

**Minor Comments**

3. L94-95: The phrasing is unclear. Suggest: "This study aims to characterize the mechanisms underlying hydroclimatic changes during the Last Interglacial using PMIP4 simulations, a weather-type classification, and dynamic-thermodynamic decomposition."

**Response:** Revised as suggested, the wording is now L94-95: "This study aims to characterize the mechanisms underlying hydroclimatic changes during the Last Interglacial."

4. L105: Replace "from" with "contributing to." Note that GCM stands for General Circulation Model, not Global.

**Response:** Revised as suggested, the wording is now L105: "General Circulation Model contributing to.." We further reviewed the manuscript to ensure that this definition is used correctly and consistently throughout.

5. L111: Clarify the data usage rationale. For example: "Due to data availability at daily resolution, only AWI-ESM and EC-ESM are used for daily-scale analyses, while all PMIP4 models are used for monthly-scale analyses."

**Response:** We have reworded the sentence for clarity. It now reads L110-115: "The analysis was based on 40-year model runs for each period, including the Last Interglacial peak at 127 ka, the Pre-Industrial (PI, 1850 CE) period, and the ERA5 reanalysis covering 1980–2020. Daily-resolution data were available only for the AWI-ESM and EC-Earth models, which were therefore used for the weather-type classification described in Section 2.2. All available PMIP4 models were used at monthly resolution to evaluate the large-scale moisture balance (see Sect. 2.3), as this temporal resolution was adequate for the study objectives."

6. L114: Should read "We evaluated each model's ability".

**Response:** Rephrased as suggested by the reviewer.

7. Table 1: Consider unifying naming conventions (e.g., EC-ESM vs. EC-Earth3-LR) to avoid confusion.]

**Response:** Thank you for this suggestion. We choose to retain the exact names of the model runs used, as some institutions provide multiple simulations. Using the full official names helps clarify precisely which model run was employed in our analysis.

8. L123-125: Explicitly state the temporal resolution (daily).

**Response:** We rephrased the sentence, see response 1.

9. L140: Why is AWI-ESM used instead of EC-ESM, given EC-ESM's higher resolution? Clarify the reasoning. L140: Clarify "alignment with proxy data". Does this mean precipitation, evaporation, circulation patterns, or simply temperatures?

**Response**: We updated the paragraph to explicitly state that the dynamic—thermodynamic decomposition was performed for all PMIP4 models listed in Table 1, rather than exclusively for AWI-ESM. The sentence has been rewritten to make this clear and to prevent misinterpretation, which now reads: "This decomposition was applied to all nine PMIP4 models listed in Table 1 to ensure a robust assessment of the moisture-balance changes across the ensemble."

10. L186: Clarify whether "increased rainfall intensity" refers to the Last Interglacial relative to present.

**Response:** We rephrased the sentence to improve clarity. It now reads L199: "consistent with proxy evidence indicating higher rainfall intensity during the Last Interglacial compared to the Pre-Industrial and present periods, as well as enhanced weathering in the Red Sea and Dead Sea regions driven by tropical systems (Palchan et al., 2018; Kiro et al., 2020)."

11. L187: Expand explanation of EC-ESM's behavior. If this discrepancy informs its exclusion from later analysis, state this explicitly earlier.

**Response:** We present our analysis of the differences between the models, following the results for both precipitation and weather types. The relevant paragraph starts in L 199: "In this section, we also clarify that our objective was to examine the proxy-based hypothesis indicating higher rainfall intensity in the southern Levant. Therefore, we decided to continue the analysis using the AWI-ESM model, which best reproduced the patterns inferred from the proxy records".

12. L193: Specify "90% of summer days".

**Response:** We added the emphasis on summer days to clarify that this is a summer system, L213.

13. When introducing section 2.3, keep the description general and explain model selection later to avoid confusion.

**Response:** We revised the introductory paragraph of this section to enhance clarity and avoid potential confusion. It now reads as follows L143:

"The increase in wet conditions reflects a shift in water flux. The net water flux, defined as precipitation (P) minus evaporation (E) over a given surface, plays a fundamental role in the water cycle. The globally averaged P minus E should theoretically balance to zero in current and future climates, assuming a closed system. However, regional variations can arise due to dynamic and thermodynamic processes influenced by climate change.

To investigate these processes, we applied a well-established decomposition framework following [Seager et al, 2010], which separates the dynamic and thermodynamic components of the moisture balance between the Last Interglacial peak and the Pre-Industrial period. In this framework, holding the humidity field

constant isolates variations attributed to changes in the dynamic component, while modifications in the humidity field, with a fixed wind field, reflect adjustments in the thermodynamic component [Seager et al., 2010; Seager et al., 2019, Elbaum et al., 2022]. This decomposition was applied to all nine PMIP4 models listed in Table 1 to ensure a robust assessment of the moisture-balance changes across the ensemble."

14. Add a reference for "enhanced contributions from the south."

**Response:** We added two references L245:

Palchan, D., Stein, M., Goldstein, S. L., Almogi-Labin, A., Tirosh, O., & Erel, Y. (2018). Synoptic conditions of fine-particle transport to the last interglacial Red Sea-Dead Sea from Nd-Sr compositions of sediment cores. Quaternary Science Reviews, 179, 123-136

. Kiro, Y., Goldstein, S. L., Kushnir, Y., Olson, J. M., Bolge, L., Lazar, B., & Stein, M. (2020). Droughts, flooding events, and shifts in water sources and seasonality characterize last interglacial Levant climate. Quaternary Science Reviews, 248, 106546.

15. Clarify whether the reported 4-7°C temperature rise is based on Table 1 model outputs or previous literature, and cite accordingly.

**Response:** We added citations and a reference to Table 1 to indicate that this statement is based on both our model outputs and previous literature. It is worth noting that the models presented in Table 1 are the same ones used in the cited studies. The sentence now reads L266-268

"Across different PMIP4 models presented in Table 1, summer temperatures in the Levant rose by more than 4°C to 7°C, consistent with previous studies [Otto-Bliesner et al. 2013, 2021].

In contrast, winter temperatures exhibited a cooling trend [Otto-Bliesner et al., 2021, 2013].

16. L262-263: Provide a reference for moisture intrusions during MIS5e.

**Response:** We added two references, in L278.

Kutzbach, J. E., Guan, J., He, F., Cohen, A. S., Orland, I. J., & Chen, G. (2020). African climate response to orbital and glacial forcing in 140,000-y simulation with implications for early modern human environments. Proceedings of the National Academy of Sciences, 117(5), 2255-2264.

Kushnir, Y., Stein, M., Biasutti, M., Kiro, Y., Goldsmith, Y., & Goldstein, S. L. (2024). Paleo aridity in the Levant driven by a strong North Atlantic latitudinal surface temperature gradient and present-day relevance. Proceedings of the National Academy of Sciences, 121(47), e2407166121.

**Reviewer 2:**

This manuscript investigates hydroclimatic variability in the Levant during the Last Interglacial peak (MIS 5e) using PMIP4 paleoclimate model simulations in combination with a synoptic weather-type classification and moisture balance decomposition. The study mostly uses the AWI-ESM model to examine precipitation characteristics and the relative roles of thermodynamic and dynamic processes. Results indicate enhanced precipitation—particularly during Red Sea Trough and Cyprus Low events—driven largely by thermodynamic factors. The authors present integration of model and proxy data but the manuscript requires major revisions to address issues of clarity, internal consistency, and the robustness of methodological and interpretative claims. The manuscript is interesting and deseves publication in Climate of the Past, provided the authors address the comments and improve its quality.

**Response:** Thank you very much for taking the time to read and review our work. We are pleased that you found the article interesting and worthy of publication. We greatly appreciate your thoughtful comments and constructive suggestions, which have helped us clarify and improve the manuscript, making it more accessible to the scientific community.

**Comments**

**1. Line 47-50:**

- a. First claim: "The Levant experienced a relatively dry climate characterized by shorter, more intense rainfall events..."

  This statement describes the entire Levant as relatively dry.
- b. Second claim: "Proxy-based reconstructions have indicated that the southern Levant experienced relatively wet conditions during this period..."

This says that the southern Levant was relatively wet.

**Conflict:**

These two statements describe opposite hydroclimatic conditions (dry vs. wet) for the same region and period, unless the author means to emphasize spatial variability within the Levant (north vs. south, for instance).

So unless the text explicitly clarifies that the southern Levant differed from the rest of the Levant, the passage reads as internally inconsistent — the same time period is characterized both as relatively dry overall and relatively wet locally.

To fix it, you could clarify the regional contrast explicitly, for example: "While the Levant as a whole experienced relatively dry conditions, proxy records suggest that the southern Levant may have been comparatively wetter during this period."

**Response**: Thank you for bringing this to our attention. Our intention in that sentence was to describe the general climatic conditions of the Last Interglacial (LIG) and then refer specifically to its peak. According to proxy evidence, the LIG as a whole was characterized by relatively dry conditions in the Levant. However, during the LIG peak, a relatively short shift towards wet conditions took place in the southern Levant.

The revised text now reads L47-50: "The Last Interglacial (approximately 130-80 ka was characterized by elevated global average temperatures, higher sea levels, and increased atmospheric CO2 concentrations compared to glacial periods [Otto-Bliesner et al., 2021; Dutton and Lambeck, 2012; Govin et al., 2015; Jouzel et al., 2007]. During the Last Interglacial (LIG), the Levant experienced an overall hyperarid climate. [Torfstein et al., 2015, 2013; Kushnir et al., 2024]. proxy-based reconstructions

have indicated that the southern Levant experienced relatively wet conditions characterized by shorter, more intense rainfall events during the Last Interglacial peak (approximately 127-122 ka), contrasting with the generally arid climate prevalent throughout much of the Last Interglacial."

- 2. Line 111: 'The analysis was based on 40-year model runs for each period.' Which period, please mention.
  - **Response**: We added a clarification indicating which periods are being referred to. It now reads L111-112: "The analysis was based on 40-year model runs for each period: the Last Interglacial peak at 127 ka, the Pre-Industrial period(PI, 1850 CE), and the ERA5 reanalysis (1980-2020)."
- 3. In Table 1, please mention spatial resolution of each model. Please also mention for what time period the simulations are available. In Table 1 you listed 9 models, and then you mostly use only two models for your analysis, what is the justification for this?

  Response: We added the spatial resolution for each model in Table 1. The reason for using only two models extensively in our analysis is that daily-resolution data were available only for the GPH variable in these models. For the remaining models, daily GPH data were not available, which prevented us from performing the weather-type classification using our method. However, we conducted the P–E analysis using monthly-resolution data, for which a larger number of models were available. For additional details, refer to Response 2 of Reviewer 1
- 4. Line 122: 'Sharav Lows', is it Sahara Lows.

  Response: We refer to the North African depression system and use the term Sharav Lows as defined in previous studies (Alpert and Ziv, 1989; Alpert et al, 2004a; Hochman et al., 2018 etc.), which differs from the more southern Sahara Lows.
- 5. Lines: 126-127: 'We compared the average Euclidean distances for the different periods to assess if the weather types during the Last Interglacial peak changed or were similar to today's.' What do you mean here by different pereiods, please

mention the periods to make it clear. What length of interglacial period you have compared with ERA5, and preindustrial.

Response 6: Please see Comment 1 from Reviewer 1

6. Line 128: What preindustrial period you have chosen, please mention.

**Response 7:** We used Pre-Industrial (PI) runs from each of the models listed in Table 1, corresponding to the atmospheric. In this part of the classification, we used only the EC-Earth and AWI-ESM models because the daily-resolution data required for the classification were available only for these two models. As detailed in the Data section, each period covers 40 years.

7. Line 140: the authors mention here the proxy data, the proxy data should be discussed in the data section before mentioning here. It is not clear which proxy is being used, and for what time period.

Response: Please see Comment 14 from Reviewer 1, we revised this paragraph to remove any discussion of proxy data at this stage. The paragraph now focuses solely on describing the decomposition framework applied to analyze the dynamic and thermodynamic components of the moisture balance between the Last Interglacial peak and the Pre-Industrial period. It now reads L147-149:

"To investigate these processes, we applied a well-established decomposition framework following Seager et al. (2010), which separates the dynamic and thermodynamic components of the moisture balance between the Last Interglacial peak and the Pre-Industrial period."

8. Fig. 2: Explain how you have appllied the bootstrap test in methods section. In sub-figure titles write the complete model name not just AWI, or EC, othewise mention these abrevations in the main text before using in the figures. In method section for clarity to the reader, please justify why you subtract interglacial period from the preindsutrial. If possible please demarkate Levant basin in Fig. 2.

**Response:** We added explanation in the Methods section describing the bootstrap approach and how it was applied.

It now reads L120-123: "To assess the statistical significance of differences in precipitation between the Last Interglacial (LIG) and Pre-Industrial (PI) periods, we applied a non-parametric bootstrap resampling test [Tibshirani and Efron, 1993]. In this approach, repeated random resampling with replacement was performed 1000 times within each period to generate a new empirical distribution of mean precipitation differences based on the original data. Statistical significance was determined at the 95% level, with differences considered significant when zero was outside the 95% confidence interval."

We also clarified the titles of the AWI and EC models. The LIG-PI difference was presented instead of the individual periods to highlight the magnitude and spatial pattern of the precipitation change between the two periods.

9. Line 172: 'First, we evaluated the PMIP4 models to assess their reliability compared to proxy-based reconstructions (see Sect. 1.3).' I do not see any evaluation of PMIP4 models in Sect. 1.3. Response: The purpose of this evaluation was not to conduct a full model performance assessment, but rather to examine whether the seasonal frequency of the simulated weather types corresponds to the well-known observed seasonal patterns, namely, the dominance of Cyprus Lows during winter and Persian Troughs during summer, and whether these patterns are consistent with the precipitation seasonality indicated by proxy-based reconstructions. We clarified this point in the text to avoid confusion L183-185.

"First, we evaluated the PMIP4 models to assess whether the simulated seasonal frequency of weather types corresponds to the observed climatological patterns with Cyprus Lows (CL) dominating in winter and Persian Troughs (PT) prevailing in summer, and to the precipitation seasonality inferred from proxy-based reconstructions (see Sect. 1.2 and 1.3)."

10.Lines 175-177: 'AWI-ESM suggests wetter winters in the Levant basin compared to the Pre-Industrial period, consistent with speleothem evidence from the Negev [Vaks et al. (2007)]. In contrast, EC-ESM shows wetter winters confined to the northern

Levant.' This statatemt is not consistent with Fig. 2b (winter) as I see similar wetter condtions for both models (AWI, and EC).

**Response 11**: We rephrased the sentence to reflect better the patterns shown in the figure." Both models exhibit a comparable increase in winter precipitation relative to the Pre-Industrial period, with the most pronounced changes occurring in the northwestern Levant."

**11.Lines 180-182: 'Autumn**

patterns also vary: AWI-ESM indicates drying in the northern Levant and increased precipitation in the south, while EC-ESM suggests widespread precipitation increases across the northern region.'

a. I do not find this statement consistent with Fig. 2A and 2E. Better demark Levant in the figure. It is not clear what you consider the Levant region, mention clearly in methods section.

**Response**: We have refined the sentence to provide a clearer description of the figure L194-195. "Whereas EC-ESM shows precipitation increases limited to the northern Mediterranean coasts, while the rest of the region remains relatively dry."

b. Fig. 3: What the y-axis shows? In the main text, please, elaborate the sub-figure that comes first, not just randomly. Not clear what periods you have chosen for ERA5, preindsutrial, and interglacial, mention them also in the main text in methods section.

**Response:** We revised the order of presentation in the text to match Figure 3 and clarified in the main text what the Y-axis represents. Now it reads L203-213: "We evaluated weather-type frequencies to explore potential drivers of precipitation differences between three periods. The y-axis in Fig. 3 represents the frequency of occurrence in percentages of each weather type, expressed as the percentage of days within each season. Both models effectively capture the seasonal-scale frequencies of weather types compared to ERA5 (Fig. 3 A—H). We compared the frequency of weather types between the Last Interglacial

(LIG) and Pre-Industrial (PI) periods to identify potential differences in their seasonal occurrence patterns (Fig. 3). In autumn, both models show an increase in Red Sea Trough frequency during the LIG compared to the PI. However, the signal is stronger in EC-ESM (Fig. 3A, E). During winter, the frequency of Cyprus Lows in AWI-ESM increases from about 40% of winter days in the PI to roughly 5% in the LIG, while EC-ESM shows no substantial changes (Fig. 3 B, F). In spring, neither model displays statistically significant differences in weather-type frequencies between the two periods (Fig. 3 C, G). During summer, both models show no changes in known rain-bearing systems; however, the Persian Trough becomes more dominant in AWI-ESM, affecting over 90 % of summer days during the LIG (Fig. 3D, H)."

Additionally, we revised the Methods section to specify the exact time span covered for each period. These changes address this comment as well as Comment 8 above and Comment 14 from Reviewer 1.

12.Line 189: 'We evaluated weather-type frequencies to explore potential drivers of precipitation differences between periods'. Which periods, not clear?

**Response:** We have revised the paragraph. Please see response 11b

13.Line 190-192: 'In AWI-ESM, the frequency of Cyprus Lows increases during winter, occurring on about 50% of winter days. At the same time, no significant changes are observed in other weather- types during autumn or spring (Fig. 3 B).' What do you mean by changes/inrease here, is it the change in frequency, if so, then compared to what, it is very confusing. While refering to results from the figures, please mention the sub-figure letter (e.g., Fig. 3A etc.) in the main text.

**Response 14:** We have re-edited the paragraph. Please see response 11b.

14.Lines: 204-206: 'Analysis of precipitation by weather type suggests a 17.3% increase in the daily average precipitation during Cyprus Low days in the AWI-ESM model compared to the Pre-Industrial period, with the most pronounced increase observed in the northern Levant, particularly over Turkey (Fig. 4A).' Isnt there also significant increase over parts of Mediterranean Sea.

**Response 15:** In this figure, only statistically significant differences are displayed. Blue regions represent areas where daily precipitation increased on both RST and CL days. The paragraph highlights the regions with the largest absolute changes, which also coincide with areas of maximum precipitation.

15.Line 220: 'At first glance, Figure 4 contrasts proxy-based findings,'. Which proxy based findings, it is not clear.

**Response 16:** We were referring to the proxy-based findings reviewed in the Introduction, not to a direct analysis of proxy data. We revised the text to make this more straightforward. The paragraph now reads L238-239:

"At first glance, Figure 4 may appear to differ from the proxy-based findings described in the Introduction, which indicate a general increase in average and high-percentile precipitation events across the Mediterranean region."

16.Lines 224-224: 'Proxy-based studies have frequently highlighted increased precipitation from southern sources rather than Mediterranean ones.' Which proxy based studies highlighted this, please cite them here.

Response 17: see Comment 14 from Reviewer 1.

17. Fig. 5 should be a table, not a figure. Further, the sub-figure numbers (e.g., 5A, 5C etc) are not marked. In section 3.3 you are describing results with reference to the figure. please refer to the figure in the very first lines (236-239).

**Response 18**: The table (formerly Figure 5) refers to the analysis of weather-type frequencies, not to the moisture-balance section (Section 3.3), which discusses a different aspect of the study. We have therefore revised the title of the table for clarity and added explicit references to it in the relevant section. Section 3.3 now

- opens with a reference to the moisture-balance analysis, which is not related to this table.
- 18.Lines 237-239: 'During winter and spring, the moisture balance remains largely unchanged across most of the Levant, with some localized decreases, particularly over modern Israel.' I this statement realy consistent with the Fig. 6 (DJF, MAM) when you say that colored regions show sinificant changes at 5% level, thus what I understand from this the changes, even thoug small, are still significant.
  - **Response 19**: You are right, all colored areas represent statistically significant changes. We have added bold formatting in the text to emphasize this. However, although the changes in winter and spring are statistically significant, their magnitudes are smaller compared to those observed in autumn and summer.
- 19.Line 280: 'Using proxy-based paleoclimate reconstructions and climate models ...' I do not see usage of proxy-based paleoclimate reconstructions in this manuscript, please clarify, how you used it.
  - **Response 20**: We did not directly analyze proxy data in this study. We intended to refer to previous proxy-based reconstructions that provided the paleoclimate context and motivation for our model-based analysis. To clarify this, we reworded the sentence as follows L301-302: "Drawing on previous proxy-based reconstructions and climate model simulations."